# Observation of bound state self-interaction in a nano-eV atom collider

Ryan Thomas [1], Matthew Chilcott[1], Eite Tiesinga[2], Amita B. Deb[1] & Niels Kjærgaard [1]

Quantum mechanical scattering resonances for colliding particles occur when a continuum scattering state couples to a discrete bound state between them. The coupling also causes the bound state to interact with itself via the continuum and leads to a shift in the bound state energy, but, lacking knowledge of the bare bound state energy, measuring this self-energy via the resonance position has remained elusive. Here, we report on the direct observation of self-interaction by using a nano-eV atom collider to track the position of a magnetically-tunable Feshbach resonance through a parameter space spanned by energy and magnetic field. Our system of potassium and rubidium atoms displays a strongly non-monotonic resonance trajectory with an exceptionally large self-interaction energy arising from an interplay between the Feshbach bound state and a different, virtual bound state at a fixed energy near threshold.

[1] Department of Physics, QSO—Centre for Quantum Science, and Dodd-Walls Centre for Photonic and Quantum Technologies, University of Otago, 730 Cumberland Street, Dunedin 9016, New Zealand. [2] Joint Quantum Institute and Centre for Quantum Information and Computer Science, National Institute of Standards and Technology and University of Maryland, Gaithersburg, MD 20899, USA. Correspondence and requests for materials should be addressed to N.K. (email: niels.kjaergaard@otago.ac.nz)

Scattering resonances are an important feature of quantum mechanics and arise whenever asymptotically free particles are coupled to an unstable bound state of the system. While the underlying mechanism of the quasi-bound formation depends on the system, its energy and lifetime can generally be determined by measurements of the position of the scattering resonance and its width. In nearly all systems, coupling between the asymptotically free states and the bound state affects the observable energy of the quasi-bound state. In the non-relativistic theory, a pair of asymptotically free particles, A and B, colliding at a kinetic energy $E$ have both background and resonant contributions to their scattering amplitude $f(E)$. Figure 1a represents diagrammatically the scattering of particles A and B at a resonance. The background scattering amplitude, $f_{bg}(E)$, is added to the resonant scattering amplitude, $f_{res}(E)$, which is an infinite sum of contributions from different order processes that start and end with asymptotically free particles A and B. When only the leading order, zero-loop term is present then $E_{AB}(E) = E_0$ is the bound state energy in the absence of coupling. However, if we include all loop orders then the quasi-bound state's energy becomes[1]

$$E_{AB}(E) = E_0 + \underbrace{\langle AB| \hat{W} \hat{G}_{bg}(E) \hat{W}^\dagger |AB\rangle}_{\delta E(E)}. \quad (1)$$

The interpretation of the energy shift $\delta E$ is that the bare bound state $|AB\rangle$ dissociates into free particles by the action of the coupling operator $\hat{W}^\dagger$, the individual particles propagate according to the background Green's operator $\hat{G}_{bg}(E)$, after which they associate back into the bound state by $\hat{W}$. This process is similar to self-energy effects that alter the mass of force-carrying particles, such as the $Z^0$ boson (Fig. 1b) in relativistic electron–positron interactions[2]. In all cases, when the bare energy of the intermediate state is fixed we can only measure the re-normalised, dressed bound state energy $E_{AB}$: it is impossible to observe the actual self-energy correction.

In atomic systems one can often change the energy of a molecular bound state using a magnetic field which leads to magnetic Feshbach resonances in the collisions of atoms[3–6]. Specifically, through the application of a magnetic field $B$, the energy of the bare (uncoupled) bound state varies according to $E_0 = \delta\mu(B - B_c)$, where $\delta\mu$ is the difference in magnetic moments between the free atoms and the molecular state AB, and $B_c$ is the field at which the bound state energy is equal to the background channel's threshold energy. This tunability has made Feshbach resonances pivotal in modern atomic physics, as one can tailor the interactions for ultracold and quantum degenerate gases[7], but it also means that, with few exceptions[8–12], the study of Feshbach resonances in ultracold gases has invariably been conducted on trapped samples, where the atomic collision energy is defined by the sample temperature. As temperatures are sufficiently low for collisions to remain in the threshold regime, such experiments cannot provide insight into self-energy effects. Instead, one must probe the resonance over a range of both collision energy and magnetic field.

In this study, we use the tunability of a magnetic Feshbach resonance in atomic collisions to directly observe the effect of self-energy corrections to the quasi-bound state energy. We detect these corrections as a non-monotonic and non-linear change of the resonance energy with respect to changes in the magnetic field as expected for an atomic system with strong coupling and a virtual bound state near threshold[13]. We find that the resonance position is well-described by a coupled-channels model.

## Results

**Scattering amplitude of an s-wave Feshbach resonance.** The background and resonant part of the elastic scattering process represented by Fig. 1 combine to give rise to a scattering amplitude expressed as[14,15]

$$f(E, \Omega) = \underbrace{\frac{(e^{2i\delta_{bg}(E)} - 1)}{2ip/\hbar} + f_{\ell>0}(E, \Omega)}_{f_{bg}(E, \Omega)}$$
$$+ \underbrace{\left( -\frac{e^{2i\delta_{bg}(E)}}{2p/\hbar} \frac{\Gamma(E)}{E - E_{AB}(E) + i\Gamma(E)/2} \right)}_{f_{res}(E)}, \quad (2)$$

where $p = \sqrt{2mE}$ is the magnitude of the relative momentum of the two particles with reduced mass $m$, $\delta_{bg}(E)$ is the s-wave background scattering phase shift, $\Gamma(E)$ is an energy-dependent resonance width, and $E_{AB}(E)$ is the energy of the quasi-bound state AB. Higher angular momentum scattering amplitudes are included in the direction-dependent term $f_{\ell>0}(E, \Omega)$ with $\Omega = (\theta, \phi)$ the direction in spherical polar coordinates. Substituting $E_0 = \delta\mu(B - B_c)$ into Eq. (1), and the result into Eq. (2), the resonant scattering amplitude becomes

$$f_{res}(E, B) = \frac{e^{2i\delta_{bg}(E)}}{2p/\hbar} \frac{\Gamma_B(E)}{B - B_{res}(E) - i\Gamma_B(E)/2}, \quad (3)$$

where $\Gamma_B(E) = \Gamma(E)/\delta\mu$, and

$$B_{res}(E) = B_c + [E - \delta E(E)]/\delta\mu, \quad (4)$$

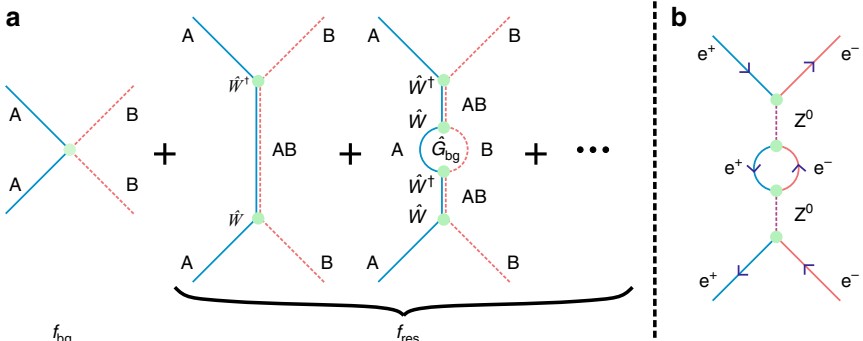

**Fig. 1** Feynman diagrams for resonant collisions. **a** The total scattering amplitude $f(E)$ for asymptotically free particles A and B is the sum of the background contribution $f_{bg}$ and the resonant contribution $f_{res}$. The lowest order contribution to $f_{res}$ has A and B forming the quasi-bound state AB which then decays back to the individual particles. The next order adds a self-energy correction to the bound state energy. AB decays into particles A and B which reform into the quasi-bound state AB. **b** Analogous self-energy correction for the mass of the $Z^0$ boson in the elastic scattering of electrons and positrons

is the resonant magnetic field. By measuring $B_{res}(E)$ at sufficiently high energies, we can infer both $\delta\mu$ and $B_c$, since $\delta E(E) \xrightarrow{E \to \infty} 0$[16], and hence determine the self-energy of the quasi-bound state.

**System under study**. We explore the collision energy and magnetic field dependence of an s-wave Feshbach resonance in $^{40}K + ^{87}Rb$ using an optical collider[12,17,18]. Specifically, we address a resonance that occurs between $^{87}Rb$ atoms in the $|F=1, m_F=1\rangle$ state and $^{40}K$ atoms in the $\left|\frac{9}{2}, -\frac{9}{2}\right\rangle$ state near 546 G ($10^4$ G = 1 T)[19–22]. As these states are the absolute ground states of $^{40}K$ and $^{87}Rb$ all collisions are elastic, and the cross section at a collision energy $E$ near the resonance can be written as[1,6]

$$\sigma(E,B) = \int |f(E,\Omega)|^2 d\Omega$$
$$= \frac{4\pi\hbar^2}{mE} \sin^2\left[\delta_{bg}(E) + \tan^{-1}\left(\frac{\Gamma_B(E)/2}{B - B_{res}(E)}\right)\right] \quad (5)$$
$$+ \sigma_{\ell>0}(E),$$

where the first term is a Beutler–Fano profile[1,5,23] as a function of $B$ for a given energy $E$, and $\sigma_{\ell>0}(E)$ accounts for the non-zero contribution to the cross section from higher ($\ell > 0$) partial waves which become important in the $^{40}K + ^{87}Rb$ system when $E/k > 100\,\mu K$, where $k$ is the Boltzmann constant. The magnetic field dependence of the cross section is experimentally determined through measurements of the fraction of atoms scattered in a collision of two cold atomic clouds

$$S(E,B) = \frac{\alpha(E)\sigma(E,B)}{1 + \alpha(E)\sigma(E,B)}, \quad (6)$$

where $\alpha(E)$ is a scaling factor determined by the peak densities of the clouds and their transverse spatial overlap at the time of collision (see Supplementary Note 1). By measuring $S(E,B)$ as a function of $B$ for fixed $E$, we determine $\delta_{bg}(E)$, $\Gamma_B(E)$, and $B_{res}(E)$.

**Experimental procedure**. We begin our experiment by loading an ultracold mixture of $^{40}K$ atoms in the $\left|\frac{9}{2}, \frac{9}{2}\right\rangle$ state and $^{87}Rb$ atoms in the $|2, 2\rangle$ state into a double-well far off-resonant optical dipole trap. The double well is formed by the intersection of a horizontal laser beam with two vertical laser beams generated by rapidly switching between two frequency pairs that drive a two-axis acousto-optic deflector[18,24]. The initial well separation is 80 μm, or approximately twice the vertical laser beam waist. We transfer Rb atoms to the $|1,1\rangle$ state and separate the double well in the presence of a strong magnetic field gradient which preferentially pushes Rb atoms into one well and K atoms into the other. The two wells are moved to final positions $z_K = 2.0$ mm for K and $z_{Rb} = -(m_K/m_{Rb})z_K = -0.92$ mm for Rb, where $m_i$ are the masses of $^{40}K$ and $^{87}Rb$, respectively. We prepare the internal states of each species of atoms using microwave and radio-frequency transitions between hyperfine states at a magnetic field of ~9 G, and we purge the wells of unwanted states and species using resonant light pulses. At the end of the state preparation, we have $3.0(3) \times 10^5$ atoms of each species in their respective wells at temperatures of $1.1(1)\,\mu K$ for K and $0.8(1)\,\mu K$ for Rb, as determined by time-of-flight absorption images. Throughout this work, numbers in parentheses correspond to the one standard deviation uncertainty combining statistical and systematic contributions. With the atoms in the desired internal quantum states, we use a dedicated, water-cooled coil pair in the Helmholtz configuration to create a stable and homogeneous magnetic field around the 546 G Feshbach resonance. The current in these coils is regulated to a fractional stability of $<10^{-5}$, and the magnetic

field is calibrated using Rabi spectroscopy between the $^{87}Rb$ $|2, 0\rangle$ and $|1, 0\rangle$ states to an accuracy of 5 mG.

Figure 2 shows the operation (see also Supplementary Movie 1) of our optical collider. From their initial positions in Fig. 2a, the atomic clouds of Rb and K are accelerated towards each other along the horizontal guide beam by steering the vertical beams of the crossed traps as illustrated in Fig. 2b. The traps are moved at velocities $v_{Rb}$ and $v_K$ such that $v_{Rb}/v_K = -m_K/m_{Rb}$. This keeps the centre-of-mass (COM) of the collisional partners at rest in the laboratory frame and ensures that expanding spheres of scattered particles are centred around a common point. Just before the collision we turn off all laser beams (Fig. 2c), so that the atoms collide in free space (Fig. 2d). After the collision, we switch the magnetic field off (see Methods) and wait until the K cloud has moved 350 μm from the collision point. At this time a halo of scattered particles will have formed (Fig. 2e), and we record the distribution of K atoms using absorption imaging perpendicular to the collision axis as in Fig. 2g. We then wait until the Rb atoms have also moved 350 μm from the collision point (Fig. 2f) and acquire an image of the Rb atoms (Fig. 2h); a frame-transfer CCD allows us to acquire images of both species of atoms in rapid succession. We determine the collision energy by acquiring two pairs of images with a time-of-flight difference of 5 ms, measuring the distance travelled by the unscattered atoms in that time, and then calculating the kinetic energy. The uncertainty in the determination of the collision energy from both systematic and statistical sources is 2% of the value.

**Measurements**. Using our optical collider, we measure the scattered fraction of atoms $S$ from absorption images (see Methods) as a function of magnetic field with the $^{40}K + ^{87}Rb$ system prepared in states $\left|\frac{9}{2}, -\frac{9}{2}\right\rangle$ ($^{40}K$) and $|1,1\rangle$ ($^{87}Rb$) for collision energies $E/k = 10$ to $300\,\mu K$. Figure 3a presents example data acquired at collision energy $E/k = 52(1)\,\mu K$. While this energy, equivalent to 4.5 neV, is far below that of conventional particle colliders, it remains two orders of magnitudes higher than the typical energy scales for studies of Feshbach resonances in trapped ultracold atomic gases as set by the sample temperature. The fraction $S$ has a pronounced Beutler–Fano lineshape associated with an inter-species Feshbach resonance centred on a magnetic field of ~546 G. In Fig. 3b, c we present post-collision absorption images of $^{40}K$ and $^{87}Rb$ for the field values where $\sigma$ (and hence $S$) attains its maximum (Fig. 3b) and minimum (Fig. 3c). At the minimum, no discernible scattering halo is visible as a result of destructive quantum interference, where the background phase shift $\delta_{bg}(E)$ of Eq. (5) is cancelled by the resonant contribution; only outgoing clouds of unscattered particles are visible in the absorption images. For the maximum, where the interference is constructive, as well as away from resonance, a halo of isotropically scattered particles emerges. The $^{40}K$ image in Fig. 3b also reveals an additional feature located at the position of the outgoing $^{87}Rb$ cloud caused by multiple scattering (see also Supplementary Fig. 1 which shows the result of a numerical simulation elucidating the effect of multiple scattering).

**Data analysis**. We fit Eqs. (5) and (6) to our data allowing $\alpha$, $\sigma_{\ell>0}$, $\delta_{bg}$, $\Gamma_B$, and $B_{res}$ to vary as functions of energy. Figure 3a shows the fitted curve for the $E/k = 52(1)\,\mu K$ case, which is typical across our entire energy range. Figure 4a–c show our fitted values for $B_{res}$, $\Gamma_B$, and $\delta_{bg}$ as functions of energy along with predictions from a coupled-channels model based on published $^{40}K + ^{87}Rb$ interaction potentials[25] that have been adjusted to match our measured resonance positions (see Methods).

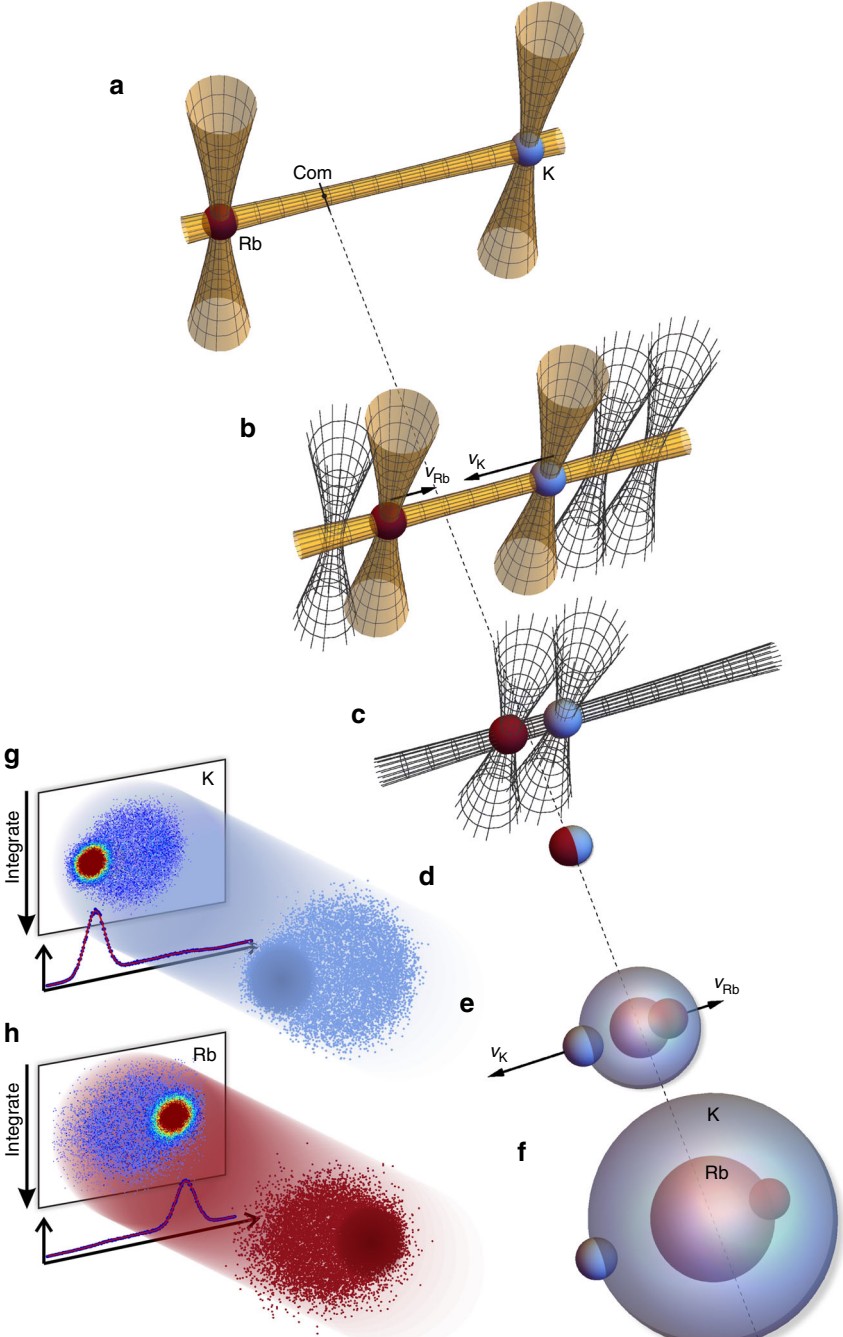

**Fig. 2** Optical collider procedure. **a** $^{40}$K (blue, right) and $^{87}$Rb (red, left) atoms, prepared in the desired internal quantum states, are held in two crossed optical dipole traps separated by ~3 mm; COM indicates the centre-of-mass for pairs of K and Rb atoms. **b** The two traps are accelerated towards each other, keeping $m_{Rb}v_{Rb} = -m_K v_K$, so that pairs of K and Rb atoms have zero total momentum on average, and their COM is at rest in the laboratory frame. **c** When the wells are separated by ≈60 μm, the optical traps are switched off. **d** The two clouds collide in free space. **e, f** The K and Rb collision halos expand at different rates. We image the K halo at the time represented in **e**, and then wait to image Rb until its halo has expanded to the equivalent size (**f**). **g, h** 3D collision halos of K (**g**) and Rb (**h**) are projected onto a 2D plane by the process of absorption imaging. We integrate the projected images in the vertical direction to obtain line densities. Line densities are fitted to extract the scattered fraction

## Discussion

A particularly striking feature of our measurements is that the resonance position $B_{res}$ is not a monotonic function of $E$ (Fig. 4a). From zero energy, $B_{res}(E)$ decreases to a minimum value near $E/k = 75$ μK before approaching a linear asymptote. This curvature is solely the result of the self-interaction energy $\delta E(E)$ due to higher-order resonant scattering processes, such as those shown in Fig. 1a. Without these processes, the resonance position

would follow the linear curve $B_c + E/\delta\mu$ in Fig. 4a. The gap between the two curves, therefore, is a direct measure of the self-energy $\delta E(E)/\delta\mu$. The marked non-monotonic behaviour, as opposed to only being non-linear, is due to the negative background scattering length which is associated with a virtual bound state just below threshold in the energetically open channel[26–28]. The effect of the virtual bound state on the open channel scattering[13,29] can also be seen in Fig. 4b,c as similarly

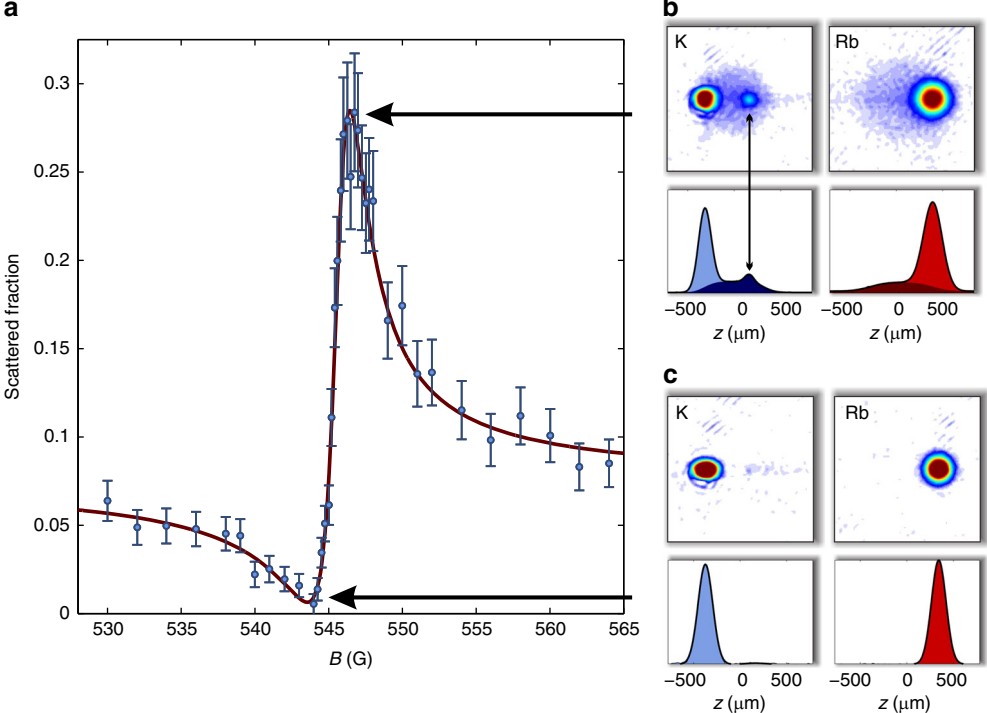

**Fig. 3** Imaging and analysis of $^{40}$K + $^{87}$Rb collisions. **a** Measured scattered fraction (circles) as a function of magnetic field for $E/k = 52(1)\,\mu$K; error bars indicate one standard deviation uncertainties. Solid line is a fit to Eqs. (5) and (6). **b**, **c** Absorption images and density profiles of K (**b**) and Rb (**c**) acquired at the magnetic fields indicated by the horizontal arrows. Light blue and light red coloured areas in density profiles indicate unscattered atoms and dark red and dark blue areas indicate scattered atoms. The double-headed arrow in **b** indicates an additional $^{40}$K feature (see text and Supplementary Fig. 1)

non-monotonic behaviours of the background phase shift and resonance width, respectively. In fact, the initial rise of $\delta_{bg}$ for $E/k < 50\,\mu$K in Fig. 4c corresponds to the Wigner threshold regime[30] and is a direct measure of the background scattering length[1], $a_{bg} = -\hbar \lim_{E \to 0} \tan\delta_{bg}(E)/p$. Here, we calculate $a_{bg} = -185.80(35)\,a_0$ from our theoretical model, which is negligibly different from the value calculated with the unmodified potentials $(-185.65a_0)$ and is consistent with previous estimates[22,31]. This non-monotonic behaviour is qualitatively different to systems with a positive background scattering length, where both the background phase shift and resonance width scale as $E^{1/2}$ for energies up to the van der Waals energy $(E_{vdW}/k \approx 600\,\mu$K for $^{40}$K + $^{87}$Rb). While measurements of molecular-binding energies also probe self-energies[22,31], we note that this is the first direct observation of such self-interaction effects in Feshbach resonances for continuum states as previous experiments worked with resonances with widths that were too small to generate appreciable curvature[10,12].

Although we have focused on one particular resonance, our technique is more general and can be applied to both isolated and overlapping Feshbach resonances. By performing our measurements at fixed collision energy, we isolate the functional dependence of both the background and resonant scattering phases and eliminate the need for a priori knowledge about how these parameters vary with energy. Elastically dominated resonances can be described by a simple Beutler–Fano function, and even resonances with significant inelastic scattering have analytic descriptions as functions of magnetic field at fixed energy[32]. The ability to study resonances with significant inelastic widths is a marked improvement over molecular-dissociation experiments, where a stable molecular state is a required starting point[8,9]. Moreover, by measuring the number of scattered atoms we avoid the need for an external interferometer to measure the scattering

phase[10]. Indeed, the number of scattered atoms can be regarded as the output of an internal interferometer, where the path length for one arm at a given energy is fixed and yields the background phase, and the path length of the other, resonant, arm can be varied with an external field: in this case, a magnetic field. Given the method's generality and the relatively high energies that we can access with our collider, we expect that this work can be extended to study Feshbach resonances in higher partial waves[33,34], where the well-defined collision axis can unambiguously determine the orbital angular momentum without observing resonance multiplets[35]. Furthermore, we can investigate the interference between Feshbach and shape resonances[9] and the effect of shape resonances on the self-energy, of which this study with an s-wave virtual state is a special case. By associating pairs of atoms into molecules, we can investigate atom–molecule and molecule–molecule collisions where there is a dearth of accurate theoretical predictions[36,37]. Finally, by changing the collisional opacity through tuning of the elastic cross section via a Feshbach resonance, we can study multiple scattering effects in jet production in unitary Fermi gases[38].

## Methods

**Counting of scattered atoms**. We count atoms by acquiring absorption images of both the rubidium and potassium clouds and converting those to density plots. The frequency of our probe laser system is not sufficiently tunable to image clouds at magnetic fields of >50 G, so we must turn off the Feshbach magnetic field prior to image acquisition. After the two atom clouds collide, we wait until the initial atom clouds are separated by 160 μm before we switch off the Feshbach magnetic field, which ensures that the magnetic field is present during the entire collision. We then wait until each cloud has travelled 350 μm from the collision point before we acquire an absorption image of the atoms; due to the differences in the speed of the Rb and K clouds, we take these images at two different times. Depending on the collision energy, the delay of K image acquisition with respect to turning off the Feshbach field coil varies between 0.5 and 4 ms. The large electrical currents (>100 A) flowing in the coil that need to be switched off, combined with metal

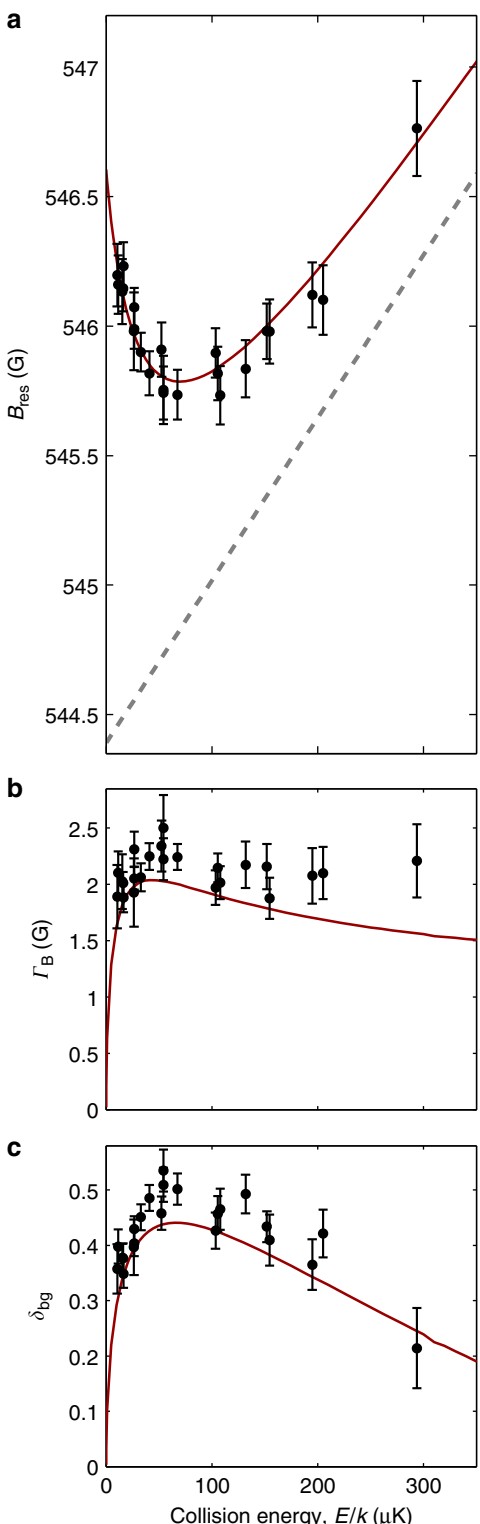

components surrounding the atoms, means that eddy currents, and hence stray magnetic fields, are present during imaging. As a result, we image in an unknown magnetic field, and while we optimise the probe laser frequency for each collision energy, the absolute number of atoms that we extract from our absorption images is not directly comparable between experiments conducted at different energies. Therefore, we calculate the fraction of atoms scattered by the collision and compare that to theoretical predictions instead of the total number of scattered atoms, as it eliminates a free parameter from our fitted model.

**Determination of scattered fraction.** We measure the scattered fraction by integrating the absorption images along an axis perpendicular to the collision axis $z$ to produce line density distributions $n_i(z)$, where $i$ can be either K or Rb. Each $n_i(z)$ can be expressed as the sum of three terms $n_i(z) = N_i^u P_i^u(z) + N_i^s P_i^s(z) + N_i^m P_i^m(z)$, where each probability density $P_i^x(z)$ is normalised to unity. The distribution $P_i^u(z)$ represents the distribution of unscattered atoms and is a Gaussian distribution. $P_i^s(z)$ describes scattered atoms whose distribution is proportional to the differential cross-section. In computing this, we neglect the initial spatial distribution and the distribution of total momentum, both of which only lead to negligible blurring of the final pattern. Finally, $P_i^m(z)$ accounts for multiply scattered atoms whose distribution cannot be described using the differential cross-section. These multiply scattered atoms are mostly K atoms due to the higher density of the Rb cloud, and the potassium atoms have a Gaussian distribution centered at the position of the Rb cloud as shown in Supplementary Fig. 1. We fit $n_i(z)$ to our integrated densities to extract the number of unscattered $N_i^u$, scattered $N_i^s$, and multiply scattered $N_i^m$ atoms. The scattered fraction $S$ is then

$$ S = \frac{\sum_i (N_i^s + N_i^m)}{\sum_i (N_i^u + N_i^s + N_i^m)}. \tag{7}$$

**Resonance parameters from theoretical modelling.** We integrate coupled-channels equations based on the Hamiltonian described in Pashov et al.[25] which includes the internal structure of the atoms in the presence of a magnetic field, the Born–Oppenheimer potentials for the $X^1\Sigma^+$ and $a^3\Sigma^+$ molecular states, the centrifugal barrier for partial wave $\ell$, and the magnetic dipole–dipole interaction. The Hamiltonian conserves the total spin projection and parity, and the weak nature of the dipole–dipole interaction means that we need only consider partial waves $\ell = 0$ and $\ell = 2$ in our calculations. Parameters are extracted by a fit of the calculated elastic cross-section to Eq. (5). We modify the long-range dispersion forces $C_n r^{-n}$ for $n = 6$, 8, and 10 by multiplying each term by retardation correction factors $f_n(r)$ obtained from the and K and Rb values $f_n^K(r)$ and $f_n^{Rb}(r)$ from ref. [39] as

$$ f_n(r) = \frac{1}{2}\left( \frac{f_n^K(\beta r)}{f_n^K(\beta R_{outer})} + \frac{f_n^{Rb}(\beta r)}{f_n^{Rb}(\beta R_{outer})} \right). \tag{8}$$

At $r = R_{outer}$, the correction factors are unity which ensures continuity of the potentials. For large $r$, $f_n(r) \propto r^{-1}$. We perform a least-squares fit to our data with $\beta$ as a free parameter, and we find $\beta = 1.000(65)$. If we instead fit to only the measurement of ref. [22], calculated at an energy of $E/k = 1$ nK, then $\beta = 0.960(19)$. These modifications have little effect on the inferred resonance width or background phase shift. For more details, see ref. [40].

## Data availability
The data that support the findings of this study are available from the corresponding author on reasonable request.

**Fig. 4** Parameters describing the $^{40}$K + $^{87}$Rb Feshbach resonance. **a–c** Resonance position $B_{res}$, width $\Gamma_B$, and background phase shift $\delta_{bg}$ as a function of collision energy $E$ for the resonance near 546 G. Solid lines are predictions from a coupled-channels model. Circles are measurements using the collider, and error bars are the one standard deviation uncertainty obtained from the fits to Eqs. (5) and (6). The dashed grey line in **a** indicates the resonance position in the absence of self-energy effects extrapolated from the asymptotic behaviour of the fitted model for $B_{res}(E)$

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

## Acknowledgements

This work was supported by the Marsden Fund of New Zealand (Contract No. UOO1121).

## Author contributions

N.K. conceived the project. R.T. performed experiments with support from A.B.D. R.T. implemented the magnetic field servo with assistance from M.C. R.T. analysed the data and conducted coupled channels calculations. E.T. provided theory. R.T. and N.K. prepared the manuscript with input and comments from all authors. N.K. supervised the project.

## Additional information

**Competing interests:** The authors declare no competing interests.

