## [Peer Review File · Nature Communications]

Reviewers' comments:

Reviewer #1 (Remarks to the Author):

The authors describe and experimentally demonstrate an elegant method for fundamental studies of two-body collisions near a magnetically-tunable Feshbach resonance, which is a topic of broad interest. In ultra-cold gases, a Feshbach resonance arises when a two-atom bound state in an energetically closed channel is hyperfine coupled to the continuum states for two asymptotically unbound (colliding) atoms in an energetically open channel. The energy difference between the open and closed channel states is magnetically tuned near resonance. By tuning the bias magnetic field at a fixed, preselected collision energy, the authors beautifully determine all of the parameters for the Beutler-Fano profile that describes the resonance, including the energy-dependent background phase shift, resonance width, and bound state energy shift.

The introduction to the paper is particularly well written for a general audience. The scattering amplitude is first decomposed into a background and resonant part, by simply rewriting the usual s-wave scattering amplitude for the full phase shift, $\delta_{bg} + \Delta_{res}$, with $\tan(\Delta_{res}) = -\Gamma/2/(E-E_{AB})$. Then the form of the energy shift of the bound state due to coupling to the continuum is nicely related to a fundamental self-energy loop, such as a Z-boson. The authors make the important point that the collision energy-dependent self-energy is not isolated in trapped atom collision experiments, since the collision energy is thermally distributed and only the average resonance energy is measured. In contrast, the present experiments measure the energy dependent resonance magnetic field to isolate the bare bound state energy from the shift.

The experiments are carried out using an optical "pipe" and two optical tweezers, where Rb atoms are trapped in one of the tweezers and K atoms are trapped in the other. The initially separated tweezers are ramped toward each other to control the collision energy, with the relative speeds chosen to keep the center of mass at rest. The atom clouds race toward each other and are released into the optical "pipe," which is extinguished just before the collision, so that scattering then occurs in free space. The resulting Rb and K halos are imaged to determine the collision fraction and the corresponding collision cross section over a range of selected energies from 10 to 300 micro-Kelvin.

The very nice data of Fig. 3 displays a classic Beutler-Fano profile for single collision energy. The authors note that background phase shift (arising from the bare continuum states) provides a reference phase for the resonant scattering phase shift, which is magnetically tunable, i.e., an "internal interferometer," as stated by the authors.

This paper is extremely well written and of great interest. It certainly should be published. I have just a one question and one comment for the authors to consider.

1) In the discussion section, the authors note that resonance position varies with energy, first decreasing and then increasing, due to the energy dependence of the bound state energy shift. The shift is directly observed as the gap between the bare bound state energy (plotted as a resonance magnetic field) and the measured resonance location. The authors state that the observed non-monotonic behavior in the shift arises from a negative background scattering length, associated with a virtual bound state "below" threshold in the open channel. I would have guessed that a (large) negative scattering length would be associated with a virtual bound state just "above" threshold in the open channel. A real bound state just below threshold would give a large positive background scattering length. Does the non-monotonic behavior arise as the virtual bound state in the open channel tunes through the bound state in the closed channel? In a simple picture, is there a coupling to both a virtual state above and a bound state below resonance?

2) First page, first paragraph: "in both cases." It was not clear to me what "both" refers to.

Reviewer #2 (Remarks to the Author):

The manuscript "Observation of bound state self-interaction in a nano-eV atom collider" is very clear and well-written. The data and agreement with theoretical modeling are impressive. To my knowledge no other experimental setup poses the degree of control and versatility required to measure over a wide and varied enough range of parameters to observe the self-interaction of a Feshbach bound state with itself via the continuum. While the result is impressive, it constitutes a benchmark of very fine details of atomic scattering theory and Feshbach resonances. It is unclear to me that this result is of sufficient interest outside the Feshbach spectroscopy community to merit publication in Nature Communications. I recommend against publication in the absence of a more thorough discussion of the consequences for and conclusions about more general scattering problems that can be drawn from this measurement and its agreement with numerical simulations. It is unclear to me how much the experimental data influenced understanding of the necessary terms and structure required to numerically simulate the bound state self-interaction. If this influence was significant, I would like to see a more detailed discussion of the understanding gained. Otherwise, a discussion of situations which this experimental technique can be applied to, but for which numerical simulations are much more uncertain, would strengthen the manuscript.

Reviewer #3 (Remarks to the Author):

The authors report on the observation of what they named "self-interaction", which is a measure of the coupling between the a Feshbach molecular state embedded and the continuum of the open-channel interaction, by using an atomic "collider", developed by the same group in Refs. [12,14,15]. This technique is capable of providing detailed information of the interaction of the collisional partners. To my understanding the self-interaction hasn't been observable in previous experiments. In the present work (for collision between Rb and K atoms), the coupling to the continuum was found to be strong due to the existence of a virtual state associated to the background interaction in the open channel. This is what make the self-interaction visible in their system. I believe these are interesting results that provides a substantial advance on the properties of molecular states embedded into the continuum.

Overall, I think the paper is very well written, with a good abstract and introduction that outlines the context of the results. The paper is written in an appropriate manner that allows non-specialists to follow the discussion. The level of precision attained in the experiment is very impressive. Therefore, I believe the present work should be accepted as a Nature Communications.

I have, however, a few comments need to be addressed by the authors before publication:

1) There is an apparent contradiction in the text. In the end of the second paragraph the authors state that "it is impossible to observe the actual self-energy correction" while in the first sentence of the third paragraph the authors state that they "directly observe self-energy corrections". I believe the authors should clarify this point. My view is that the self-energy is not a quantum mechanical observable. The only thing one can is to observe deviations from the energy of the bare molecular state and its actual energy. The energy of the bare molecular state (also not an observable) can only be provided by the theory. I believe this is an important point to clarify in the text. That is also the case for the title, where it currently can lead the reads to think that self-interaction i indeed a physical observable.

2) Below Eq. (4) the authors need to define "Wigner threshold energy".

3) It would be extremely useful if the authors could quote the value of the background scattering length obtained experimentally and compare that to theoretical calculations.

Authors' response to Referee reports on the manuscript “Observation of bound state self-interaction in a nano-eV atom collider”

Considered for Nature Communications

Summary

Referee #1 states that the “*paper is extremely well written and of great interest*” with the introduction “*particularly well written for a general audience*” and that “*It certainly should be published*”. Referee #2 finds that the manuscript “*is very clear and well-written*” but also that “*While the result is impressive, it constitutes a benchmark of very fine details of atomic scattering theory and Feshbach resonances. It is unclear to me that this result is of sufficient interest outside the Feshbach spectroscopy community to merit publication in Nature Communications*”. Finally, Referee #3 finds the paper to “*very well written in “an appropriate manner that allows non-specialists to follow the discussion and that “the present work should be accepted as a Nature Communications”*”.

Point-by-Point response to Referee’s remarks.

We are most grateful for the remarks we have received from all three referees. We detail our responses below.

Referee #1

The authors describe and experimentally demonstrate an elegant method for fundamental studies of two-body collisions near a magnetically-tunable Feshbach resonance, which is a topic of broad interest. In ultra-cold gases, a Feshbach resonance arises when a two-atom bound state in an energetically closed channel is hyperfine coupled to the continuum states for two asymptotically unbound (colliding) atoms in an energetically open channel. The energy difference between the open and closed channel states is magnetically tuned near resonance. By tuning the bias magnetic field at a fixed, preselected collision energy, the authors beautifully determine all of the parameters for the Beutler-Fano profile that describes the resonance, including the energy-dependent background phase shift, resonance width, and bound state energy shift.

The introduction to the paper is particularly well written for a general audience. The scattering amplitude is first decomposed into a background and resonant part, by simply rewriting the usual s -wave scattering amplitude for the full phase shift, $\delta_{bg} + \delta_{res}$, with $\tan(\delta_{res}) = -\Gamma/2/(E - E_{AB})$. Then the form of the energy shift of the bound state due to coupling to the continuum is nicely related to a fundamental self-energy loop, such as a Z -boson. The authors make the important point that the collision energy-dependent self-energy is not isolated in trapped atom collision experiments, since the collision energy is thermally distributed and only the average resonance energy is measured. In contrast, the present experiments measure the energy dependent resonance magnetic field to isolate the bare bound state energy from the shift.

The experiments are carried out using an optical “pipe” and two optical tweezers, where Rb atoms are trapped in one of the tweezers and K atoms are trapped in the other. The initially separated tweezers are ramped toward each other to control the collision energy, with the relative speeds chosen to keep the center of mass at rest. The atom clouds race toward each other and are released into the optical “pipe,” which is extinguished just before the collision, so that scattering then occurs in free space. The resulting Rb and

K halos are imaged to determine the collision fraction and the corresponding collision cross section over a range of selected energies from 10 to 300 micro-Kelvin.

The very nice data of Fig. 3 displays a classic Beutler-Fano profile for single collision energy. The authors note that background phase shift (arising from the bare continuum states) provides a reference phase for the resonant scattering phase shift, which is magnetically tunable, i.e., an “internal interferometer,” as stated by the authors.

This paper is extremely well written and of great interest. It certainly should be published. I have just a one question and one comment for the authors to consider.

• Referee #1, Remark (1) : In the discussion section, the authors note that resonance position varies with energy, first decreasing and then increasing, due to the energy dependence of the bound state energy shift. The shift is directly observed as the gap between the bare bound state energy (plotted as a resonance magnetic field) and the measured resonance location. The authors state that the observed non-monotonic behavior in the shift arises from a negative background scattering length, associated with a virtual bound state “below” threshold in the open channel. I would have guessed that a (large) negative scattering length would be associated with a virtual bound state just “above” threshold in the open channel. A real bound state just below threshold would give a large positive background scattering length. Does the non-monotonic behavior arise as the virtual bound state in the open channel tunes through the bound state in the closed channel? In a simple picture, is there a coupling to both a virtual state above and a bound state below resonance?

Response (1,1): We are aware that the notion of virtual levels at positive energies exists in the literature. The monograph “*Theoretical Nuclear Physics*” by Blatt and Weisskopf for example writes (when discussing neutron proton scattering):

Actually, a_s , is negative, and there is no bound singlet state. It is nevertheless customary to define an energy B_s^* in the following way: (3.22) is used to find R (which now turns out to be negative); the sign of R is reversed arbitrarily and substituted into (2.7) to find $B \equiv B_s^*$. Then there is said to be a “virtual state” at the (positive) energy B_s^* . The large absolute value of the singlet scattering length is interpreted to mean that the “virtual level” of the system is not very far above zero energy. It must be understood, however, that this is strictly a way of speaking. There is no distinction between the wave function at this particular energy $E = B^*$ and any other positive energy.

However, modern treatments would have virtual bound states showing up as poles of the (analytically continued) S-matrix defined on a complex momentum plane. Here the pole corresponding to a real bound state lies on the positive imaginary momentum axis. For $\ell = 0$ states, as the potential is weakened, the momentum moves down the imaginary axis until it crosses threshold ($k = 0$) and proceeds along the negative imaginary axis. Poles on the negative imaginary momentum axis are called virtual bound states. The complex momentum plane is mapped onto a two-sheeted Riemann energy surface. Here the real and virtual bound states both live on the *negative* real axis, the former on the first physical Riemann sheet, the latter on the second unphysical Riemann sheet. These are “virtual” bound states, and in our manuscript we state that they lie *below* threshold in the sense that their energies are negative. More details can be found in J. R. Taylor’s *Scattering Theory* (chapter 13-b) or in Landau & Lifshitz’ *Quantum Mechanics, Non-relativistic Theory, 3rd ed* (§133). Further sources include R.G. Newton’s *Scattering theory of Particles and Waves, 2nd ed*, p.357:

“If on the negative real axis, the poles of S on the second sheet are sometimes called *virtual states* and also *antibound states*”,

A.G. Sitenkov *Lectures in Scattering Theory*, p.113 :

“The virtual states have negative energy levels.”

Also, Arno Bohm’s *Quantum Mechanics: Foundations and Applications 2nd edition*, p. 475:

“Poles on the negative real axis of the unphysical sheet are called virtual-state poles. A virtual state is one that would be bound if the interaction were more attractive and a virtual state close to threshold causes a large cross section at low energy.”

Hence we think we are justified in stating that the virtual bound state is located below threshold (inasmuch it is possible to talk about the “location” of a “virtual” state). However, recognising the potential for confusion we have rephrased the pertaining section of the Discussion slightly and added three references to author’s assigning the energy of a virtual state to be negative.

• Referee #1, Remark (2) : *First page, first paragraph: “in both cases.” It was not clear to me what “both” refers to.*

Response (1,2): This was a hanging phrase left over from a previous revision (which referred to relativistic/non-relativistic). We have changed the sentence fragment ...in both cases its energy and lifetime can be determined... to **...its energy and lifetime can generally be determined...**

Referee #2

The manuscript “Observation of bound state self-interaction in a nano-eV atom collider” is very clear and well-written. The data and agreement with theoretical modeling are impressive. To my knowledge no other experimental setup posses the degree of control and versatility required to measure over a wide and varied enough range of parameters to observe the self-interaction of a Feshbach bound state with itself via the continuum. While the result is impressive, it constitutes a benchmark of very fine details of atomic scattering theory and Feshbach resonances. It is unclear to me that this result is of sufficient interest outside the Feshbach spectroscopy community to merit publication in Nature Communications. I recommend against publication in the absence of a more thorough discussion of the consequences for and conclusions about more general scattering problems that can be drawn from this measurement and its agreement with numerical simulations. It is unclear to me how much the experimental data influenced understanding of the necessary terms and structure required to numerically simulate the bound state self-interaction. If this influence was significant, I would like to see a more detailed discussion of the understanding gained. Otherwise, a discussion of situations which this experimental technique can be applied to, but for which numerical simulations are much more uncertain, would strengthen the manuscript.

Initially, we would like to point out that the the official scope of *Nature Communications* is that the “journal aims to represent important advances of significance to specialists within each field”. It could be argued that the statement of the referee

“While the result is impressive, it constitutes a benchmark of very fine details of atomic scattering theory and Feshbach resonances. It is unclear to me that this result is of sufficient interest outside the Feshbach spectroscopy community”

makes a case for exactly that.

Notwithstanding that we believe that the referee might be applying inappropriate criteria to assess the merits of our work, we disagree that our manuscript will be of interest only to the “Feshbach spectroscopy community”. Firstly, as we have stressed in the initial three paragraphs of the introduction, the concept of a self-energy correction applies to all systems experiencing resonant scattering. It is not specific to ultracold atomic physics or to magnetically tunable Feshbach resonances. Secondly, our

main result is not solely that we can provide an energy-dependent parametrization of the Feshbach resonance which can then be used to fine-tune interaction potentials, although this is a very useful result. An equally important result is that we observe non-monotonic behaviour of the resonance position (as predicted by Marcelis see ref. 28), width, and background phase shift which has not, as far as we know, been observed before. That these quantities behave as they do is a direct consequence of the s-wave virtual bound state below resonance. This is a general phenomenon and not restricted simply to ultracold atomic physics.

To address the referee's comments, we have made two main changes to the manuscript. First, we have included a section in the Methods where we describe the change we have made to the published model potentials in order to match our measurements. Second, we have expanded our discussion of the future applications of our technique. Specifically, we point out that our technique could be used to probe atom-molecule and molecule-molecule collisions, especially between polar molecules, where there is a lack of theoretical results.

Referee 3

The authors report on the observation of what they named "self-interaction", which is a measure of the coupling between the a Feshbach molecular state embedded and the continuum of the open-channel interaction, by using an atomic "collider", developed by the same group in Refs. [12,14,15]. This technique is capable of providing detailed information of the interaction of the collisional partners. To my understanding the self-interaction hasn't been observable in previous experiments. In the present work (for collision between Rb and K atoms), the coupling to the continuum was found to be strong due to the existence of a virtual state associated to the background interaction in the open channel. This is what make the self-interaction visible in their system. I believe these are interesting results that provides a substantial advance on the properties of molecular states embedded into the continuum.

Overall, I think the paper is very well written, with a good abstract and introduction that outlines the context of the results. The paper is written in an appropriate manner that allows non-specialists to follow the discussion. The level of precision attained in the experiment is very impressive. Therefore, I believe the present work should be accepted as a Nature Communications.

I have, however, a few comments need to be addressed by the authors before publication: .

- Referee #3, Remark (1) : *There is an apparent contradiction in the text. In the end of the second paragraph the authors state that "it is impossible to observe the actual self-energy correction" while in the first sentence of the third paragraph the authors state that they "directly observe self-energy corrections". I believe the authors should clarify this point. My view is that the self-energy is not a quantum mechanical observable. The only thing one can is to observe deviations from the energy of the bare molecular state and its actual energy. The energy of the bare molecular state (also not an observable) can only be provided by the theory. I believe this is an important point to clarify in the text. That is also the case for the title, where it currently can lead the reads to think that self-interaction i indeed a physical observable.*

Response (3,1): The referee is correct in pointing out that, strictly speaking, the self-energy is not a physical observable. Therefore, we cannot, in the most precise sense of the word, "observe" self-energy corrections. A more careful wording would be that we observe the *effect* of the bound state self-interaction, or that we *infer* the existence of the bound state self-interaction. We feel, however, that our usage of "observe" is consistent with its use in the literature, especially regarding the title. For instance, the papers "Observation of a new particle in the search for the Standard Model Higgs boson with the ATLAS detector at the LHC" (*Physics Letters B*, **716** (1) 1-29, 2012) and "Observation of a new boson at a mass of 125 GeV with the CMS experiment at the LHC" (*Physics Letters B*, **716** (1) 30-61, 2012) claim to "observe" a new particle when in fact they observe an excess of particular decay events from which the authors then infer the existence of a new particle based on their theoretical frame-

work. Similarly, we observe a non-linear resonance trajectory through a space of collision energy and magnetic field, and from these observations we infer the existence of self-energy corrections.

While we do not feel that our use of “observe” is outside of its currently accepted use, we agree that the referee’s specific example regarding the two back-to-back statements

...it is impossible to observe the actual self-energy correction.

In this study, we use the tunability of a magnetic Feshbach resonance in atomic collisions to directly observe the self-energy corrections to the quasi-bound state energy.

requires more clarity. Therefore, we have changed the second sentence to:

In this study, we use the tunability of a magnetic Feshbach resonance in atomic collisions to directly observe the effect of self-energy corrections to the quasi-bound state energy as a non-linear change of the resonance energy with respect to changes in the magnetic field.

- Referee #3, Remark (2) : *Below Eq. (4) the authors need to define “Wigner threshold energy”.*

Response (3,2): We have removed the reference to the Wigner threshold energy, as it has no agreed-upon definition, and have replaced the sentence fragment

...and $\sigma_{\ell>0}(E)$ accounts for the non-zero contribution to the cross section from higher ($\ell > 0$) partial waves when E is larger than the Wigner threshold energy, which is $E/k \approx 100 \mu\text{K}$ for $^{40}\text{K}+^{87}\text{Rb}$, where k is the Boltzmann constant.

with

...and $\sigma_{\ell>0}(E)$ accounts for the non-zero contribution to the cross section from higher ($\ell > 0$) partial waves which become important in the $^{40}\text{K}+^{87}\text{Rb}$ system when $E/k > 100 \mu\text{K}$, where k is the Boltzmann constant.

In the Discussion, we have added a reference to Wigner’s original paper.

- Referee #3, Remark (3) : *It would be extremely useful if the authors could quote the value of the background scattering length obtained experimentally and compare that to theoretical calculations.*

Response (3,3): We have included a value of the background scattering length as obtained from the coupled-channels model describing our data, $-185.80(35) a_0$, in the text, and we have compared this with both the value that is obtained from the potentials in Ref. 24 and to values used in previous experimental works.

REVIEWERS' COMMENTS:

Reviewer #1 (Remarks to the Author):

I believe that the authors have satisfactorily addressed my comments. In particular, they explain nicely their terminology regarding the "virtual bound state" associated with a negative scattering length having an energy just below threshold which answers my question.

The authors also have reworded their statement regarding the "observation of the self energy" to read the observation of "the effects of" the self energy, which is more precise and appears to address the other referee's criticism.

As the referees state, the manuscript is very well written. Hence, at this point, I believe that it should be accepted for publication.

Reviewer #2 (Remarks to the Author):

I withdraw my concerns about publication of this manuscript in Nature Communications. It is a solid, new result in the field of Feshbach spectroscopy, and it is well written. In light of the comments of the other referees and the authors' response, it is clear that my original objections were overly demanding of the scope of the result, and that the result itself is of broader interest than my initial assessment led me to believe. I recommend publication of this revised manuscript.

Reviewer #3 (Remarks to the Author):

I appreciate the author's response to my concerns and I believe they have properly addressed them. With that in view, and also analyzing the author's response to the other referees' comments, I would like to recommend this manuscript for publication in Nature Communications.